# Assessing the Effect of Curcumin on the Oral Mucosal Cytomorphometry and Candidal Species Specificity in Tobacco Users: A Pilot Study

**DOI:** 10.3390/healthcare10081507

**Published:** 2022-08-10

**Authors:** Prishita Mehta, Rashmi Bhavasar, Namratha A. Ajith, Rahul P. Bhavsar, Maha A. Bahammam, Mohammed Mousa H. Bakri, Khalid J. Alzahrani, Ahmad A. Alghamdi, Ibrahim F. Halawani, Shilpa Bhandi, A. Thirumal Raj, Shankargouda Patil

**Affiliations:** 1Department of Oral and Maxillofacial Pathology and Microbiology, K.M. Shah Dental College and Hospital (KMSDCH), Sumandeep Vidyapeeth, Vadodara 391760, India; 2Department of Pharmacology, Ulhas Patil Medical College and Hospital, Jalgaon 425001, India; 3Department of Periodontology, Faculty of Dentistry, King Abdulaziz University, Jeddah 21589, Saudi Arabia; 4Executive Presidency of Academic Affairs, Saudi Commission for Health Specialties, Riyadh 11614, Saudi Arabia; 5Department of Oral and Maxillofacial Surgery and Diagnostic Sciences, College of Dentistry, Jazan University, Jazan 45142, Saudi Arabia; 6Department of Clinical Laboratories Sciences, College of Applied Medical Sciences, Taif University, Taif 21944, Saudi Arabia; 7Department of Restorative Dental Sciences, Division of Operative Dentistry, College of Dentistry, Jazan University, Jazan 45142, Saudi Arabia; 8Department of Cariology, Saveetha Dental College & Hospitals, Saveetha Institute of Medical and Technical Sciences, Saveetha University, Chennai 600077, India; 9Department of Oral Pathology and Microbiology, Sri Venkateswara Dental College and Hospital, Chennai 600130, India; 10Department of Maxillofacial Surgery and Diagnostic Sciences, Division of Oral Pathology, College of Dentistry, Jazan University, Jazan 45142, Saudi Arabia; 11Centre of Molecular Medicine and Diagnostics (COMManD), Saveetha Dental College and Hospitals, Saveetha Institute of Medical and Technical Sciences, Saveetha University, Chennai 600077, India

**Keywords:** candida, chromagar, cytomorphometry, pre-cancer, tobacco

## Abstract

Objectives: Tobacco consumption is of major concern for public health. Compromised oral hygiene accentuated by tobacco leads to alteration in the oral mucosa and microbiome, including *Candida*, and its species can be identified rapidly using CHROMagar. Curcumin, a naturally available compound possesses antioxidant, anti-inflammatory, anti-microbial, anti-carcinogenic, anti-fungal, and immunomodulatory properties. Hence, a comprehensive study was planned. Aim: To evaluate and compare cytomorphometric analysis and *Candida* colonization and speciation in tobacco users before and after the use of curcumin gel. Materials and Methods: The study comprised a total of 120 participants (the study (tobacco habit) group, n = 60 and control (healthy) group, n = 60). The intervention was the application of curcumin gel over the lesion area three times daily for 2 months. All participants’ oral health status was assessed, followed by cytomorphometric analysis and *Candida* colonization and speciation using CHROMagar. Results: Cytomorphometric analysis showed statistically significant differences in the control and study group for cell diameter (CD), nuclear diameter (ND), CD:ND ratio, and micronuclei (*p* = 0.0001). *Candida* colonization had a significantly higher number of colonies in the habit group when compared to the control group. *Candida tropicalis* was predominant in the study group, whereas *Candida albicans* was predominant in the control group. In the study group, after intervention with curcumin, a statistically significant difference was seen in nuclear diameter, CD:ND ratio, and micronuclei. There was a reduction in the number of *Candida* colonies, and *Candida albicans* was the predominant species observed in the study group after the intervention of curcumin and discontinuation of habit. Conclusion: Curcumin was found to reduce the number of micronuclei and also decreased *Candida* colonization, along with the discontinuation of habit in tobacco users.

## 1. Introduction

Tobacco causes high morbidity and mortality not only in developing but also in developed countries despite great achievements in public health globally [1]. India stands out second in terms of the number of tobacco users. Tobacco preparations in India are available in both smoking and smokeless forms [2]. Tobacco-related habits may prompt the formation of clinically detectable oral lesions, such as tobacco pouch keratosis [3], smokers’ palate [4], oral submucous fibrosis [5], and leukoplakia [6], with risk of malignancy [7]. Constant contact of tobacco products with oral mucosa induces a local rise in temperature and causes inflammation, which not only inhibits many systemic immune functions, but its metabolites also cause oxidative stress on tissues. Tobacco and related substances act to the extent of causative agents for the epigenetic alteration of oral epithelial cells and other changes such as variations in nuclear sizes within the epithelial cells [8,9]. Carcinogenesis initiates when normal epithelium reveals initial cytomorphometric changes and is further distressed by genetic alterations. For cytopathology, the assessment of nuclear size as determined by nuclear diameter (ND), the cytoplasm of cell size evaluation as cell diameter (CD), and their ratio is of significance for the evaluation of potentially malignant oral lesions [8,9]. Along with cytomorphometry, genomic damage is detected in the form of micronuclei on cytopathology [10].

Micronuclei are very characteristic and act as a biomarker for chromosomal damage caused by genotoxic agents from tobacco and its related substances and can be ascertained on routine cytology by using light microscopy [2]. 

Poor oral hygiene due to tobacco product retention also leads to alteration of the oral microbiome. *Candida* exists as commensals and further habits of smokeless or smoking tobacco, along with alteration of oral mucosa, facilitate *Candida* colonization in the oral cavity [11].

*Candida,* most of the time, is a superimposed infection in tobacco chewers’ lesions and the presence of *Candida* makes lesions vulnerable to malignancy [12,13] by showing the phenotypic change from yeast to hyphae form [11] and may also occur due to its endogenous nitrosamine production [11].

Association of *Candida* was reported by studies [14] in 60% of speckled leukoplakia with cellular atypia. 

Sabouraud dextrose agar is considered to be the gold standard for the detection of *Candida* colonies, whereas conventional means of *Candida* detection are effective but very time-consuming [12]. CHROMagar not only parallels conventional methods of *Candida* identification but also makes identification of *Candida* species very rapid, simple, easy, and cost-effective [15]. It also has the advantage of suppressing bacterial growth compared to Sabouraud Dextrose Agar (SDA), and it does not affect the viability of subsequent subcultures of *Candida* colonies, even in poor resource settings [16].

Tobacco chewers’ lesions, if diagnosed early, along with discontinuation of habits, can prevent the occurrence of oral cancer. Among various therapeutic modalities available, curcumin is a naturally occurring polyphenolic compound derived from Curcuma Longa [17] and has been known for thousands of years for its antioxidant, anti-inflammatory, anti-microbial, anti-carcinogenic, anti-fungal, and immunomodulatory properties [18] and preparations are available as gel, capsules and liquid drops. Gel preparation offers better adherence and longer residence time resulting in better exposure to the drug at the site of action. Curcumin possesses its anti-carcinogenic activity as suggested by various in vitro and clinical studies either by increasing levels of vitamins C and E or preventing lipid peroxidation and DNA damage [19]. The methanol extract of turmeric demonstrated antifungal activity against *Candida albicans* [17]. Its anti-*Candida* activity was demonstrated against 38 different strains of *Candida* including some fluconazole-resistant strains and clinical isolates of *C. Albicans*, *C. glabrata*, *C. krusei, C. tropicalis*, and *C. guilliermondii* [20].

### 1.1. Need of the Study

As clinical examination alone may not be accurate in predicting the course of the disease, the comprehensive evaluation may lead to detailed insight into tobacco users’ lesions with the therapeutic effect of curcumin, if any. The literature is scarce in comprehensive evaluation of cytomorphometric analysis and *Candida* speciation in the lesions of tobacco users before and after application of curcumin gel. Hence, the present study was planned with the null hypothesis that there is no difference in cytomorphometric analysis and *Candida* speciation before and after the use of curcumin gel in tobacco users. The following were the aims and objectives of the study.

### 1.2. Aims 

To evaluate and compare cytomorphometry analysis and *Candida* colonization and speciation using CHROM agar in tobacco users before and after use of curcumin gel and a healthy control group. 

### 1.3. Objectives

To evaluate and compare cell diameter, nuclear diameter, CD:ND ratio, and micronuclei before use of curcumin gel in the study group (tobacco users) and control group (healthy participants);To evaluate and compare cell diameter, nuclear diameter, CD:ND ratio, and micronuclei before and after the use of curcumin gel in the study group;To evaluate and compare *Candida* colonization (colony forming units—CFU) and speciation prior use of curcumin gel in the study group and control groups;To evaluate and compare *Candida* colonization and speciation before and after the use of curcumin gel in the study group.

### 1.4. Methods

The study was a pilot and prospective study with intervention by curcumin gel. A total of 120 participants were recruited, comprising 60 participants in each group. Sample size was calculated based on the number of patients visiting the outpatient department of K.M. Shah Dental College and Hospital and considering a two-month duration for the study. 

The grant for the research was provided by Short Term Studentship- Indian Council of Medical Research (STS-ICMR) and in accordance with the guidelines provided by STS-ICMR, the study had to be completed within 2 months, therefore keeping the number of Out patient Department (OPD) in mind, the following sample size was decided with the help of the following formula.

Department OPD of the last 3 months was around 175. Keeping that in mind,
N = N/1 + Ne^2^
N = 175/1 + (0.05)^2^
N = 120
N = 175, e = 0.05, confidence interval = 95%. 

Total patients required, 120 including for considering loss of follow-up if any: 

**Control group (Group A):** healthy participants with no habit and normal mucosa (60 patients) 

**Study group (Group B):** tobacco abusers with the presence of lesions (60 patients)

For the present study, approval of ICMR studentship funds was followed for the institutional ethics committee approval from Sumandeep Vidyapeeth Institutional Ethics Committee, with approval number SVIEC/ON/DENT/SRP/20074, dated 1 July 2020. The study was carried out after receiving approval for Short-Term Studentship by the Indian Council of Medical Research (Ref ID: 2020-04234).

All participants were screened by the primary investigator and recruited into the study after obtaining their informed consent.

**The selection criteria for the study group** participants were as follows:

**(a) Inclusion criteria for Group A:** (Figure 1 and Figure 2)

1. Participants with ages ranging from 18 to 40 years. 

2. History of tobacco and related substance use for a minimum of 6 months with the presence of a detectable lesion in the oral cavity. 


**(b) Exclusion criteria for Group A:**


1. Participants with any systemic illness. 

2. Participants who had undergone any type of treatment for tobacco lesions in the past or present. 

3. Participants with a history of allergy to curcumin and related products. 

There are studies reporting that the duration and frequency of tobacco use [21,22] is important for developing pre-malignant lesions, but at the same time, every human body reacts differently, and, therefore, a person can develop pre-malignant lesion at an early stage or may develop it later. 

Prolonged exposure as tobacco in chewable form, especially where tobacco is kept in contact with the mucosa for a longer time, leads to leaching and concentration of carcinogens at a localized area and increases the chances of developing lesion. As reported by Garg et al. [21], in these cases, a longer individual chewing cycle, even of short duration, may be hazardous. Lesions may also develop as a protective mechanism for localized carcinogen action.

Tobacco components increase the production of reactive oxygen species, cell turnover, collagen synthesis, and also alter DNA damage.

Additionally, depending upon the damage occurring, clinical presentation varies from greyish-white ill-demarcated lesions to ulceration. Recent research by Halboub et al. and Monika et al. [22] reported that various types of smokeless tobacco harbor bacteria that play a role in carcinogenicity. Mechanical and chemical irritation from smokeless tobacco also induces melanin pigmentation.

Miller et al. [22] reported that smokeless tobacco lesions should be treated by habit discontinuity, which shows resolution within 6 weeks to 6 months, so a minimum of 6 months’ duration for history of tobacco habit was considered as inclusion criteria for the study. 

**Control Group B** consisted of participants visiting the institute for routine health check-ups and were asymptomatic, age- and gender-matched healthy participants, voluntarily ready to be a part of the study and having no history of tobacco habit in any form.

## 2. Materials and Methods

### 2.1. Clinical Examination: (Figure 3)

Each patient who reported to the OPD satisfying our inclusion criteria was a part of the study. 

Participants were provided with information about the study in the vernacular language through information sheet and signed informed consent form was obtained from all participants in the study. 

For present study, translation, back-translation in the local language of Gujarati and Hindi, and validation of the same were mandatory protocols for ethics committee approval and also were verified by all members of the ethics committee; herewith, we have attached a copy of the case history proforma for the history of habits, informed consent form, and participant information sheet for reference. 

Participants were interviewed for a detailed history of habits, especially those related to tobacco consumption, and oral health status was recorded.

Tobacco and related substance intake history included details of tobacco habits regarding the form of tobacco intake, the quantity of tobacco substance, duration, and frequency. Details of tobacco history were recorded in a questionnaire proforma and were based on self-information given by the participants. Performa had questions for months/years since participants have been consuming tobacco, whereas quantity was based on the number of packets consumed by participants. The frequency of habits was recorded as amount of tobacco chewed per day or the number of beedis/cigarettes smoked per day.

2.Thorough intra-oral and extra-oral examination were performed and a clinical diagnosis for each lesion was made [23].


**Extra-oral examination:**


Face and head were assessed to find abnormal findings such as symmetry, swelling or discoloration

Neck was palpated to assess major lymph nodes for any lumps, swelling, and tenderness 


**Intra oral examination:**


Examination was done with the help of mouth mirror and probe. Lips, palate, tongue, floor of mouth, buccal mucosa, labial mucosa, and fauces were checked for discoloration, texture, keratinization, swelling, consistency or any other abnormality, such as fibrous bands.

Lesions were diagnosed as **tobacco pouch keratosis** when the mucosa appeared gray or gray-white and almost translucent in localized areas contacting the tobacco. The surface of the mucosa appears white and is granular or wrinkled and in some cases, a folded character may be seen. The stretched mucosa appears fissured or rippled, and a “pouch” is usually present. This white tobacco pouch may become leathery or nodular in long-term, heavy users, and the lesion is usually asymptomatic [7]. The oral tissue can be white or may appear wrinkled and, in rare instances, may elicit folding. When the mucosa is stretched, it may resemble fissures, and it may present with a “pouch”. In patients with a history of consuming tobacco for a prolonged duration, the oral mucosal pouch may elicit changes including leathery characteristics, and the lesion is usually asymptomatic.

**Smoker’s palate** was diagnosed when palatal mucosa was seen as diffusely grayish or whitish and erythematous. Oral lesions may exhibit papules with a central erythematous punctate. Such lesions may form on both the soft and the hard palate, especially in long-term tobacco smokers.

The areas with clinical manifestations are those that are in constant exposure to the heat generated during smoke inhalation. 

“Reverse smoking” has been shown to elicit severe manifestations, including lesions of the erythroleukoplakic variety. Such cases are frequently noted in long-term tobacco smokers [7].

Leukoplakia was inferred in those cases which presented with white lesions which were not scrapable and were demarcated. The surface of the lesion may appear to have multiple fissures and may elicit a cracked, mud-like feel on palpation [7].

The diagnosis of, was rendered if the clinical presentation included either palpable fibrotic bands or at the least exhibited an opaque, blanched appearance. The diagnosis can also be rendered in cases wherein the patient complains of a sensation of irritation/burning following the consumption of spicy foods. The burning sensation may be accompanied by vesicular and ulcerative changes in the oral mucosa, which may progress to oral mucosal stiffness which may culminate in reduced mouth opening [7].

A healthy control group was selected from the asymptomatic participants visiting institute’s outpatient department for routine oral health check-ups and having no history of tobacco use or any systemic diseases. 

For standardization purposes, all the participants were given similar oral hygiene instructions as a part of their routine treatment protocol for the next 2 months. 

Following were the oral hygiene instructions [24,25,26]

1. Brush teeth twice a day with soft bristled toothbrush, once in morning and once at night;

2. Rinse and swish with water after every meal;

3. Floss once daily;

4. Brush the tongue. 

**Figure 3 healthcare-10-01507-f003:**
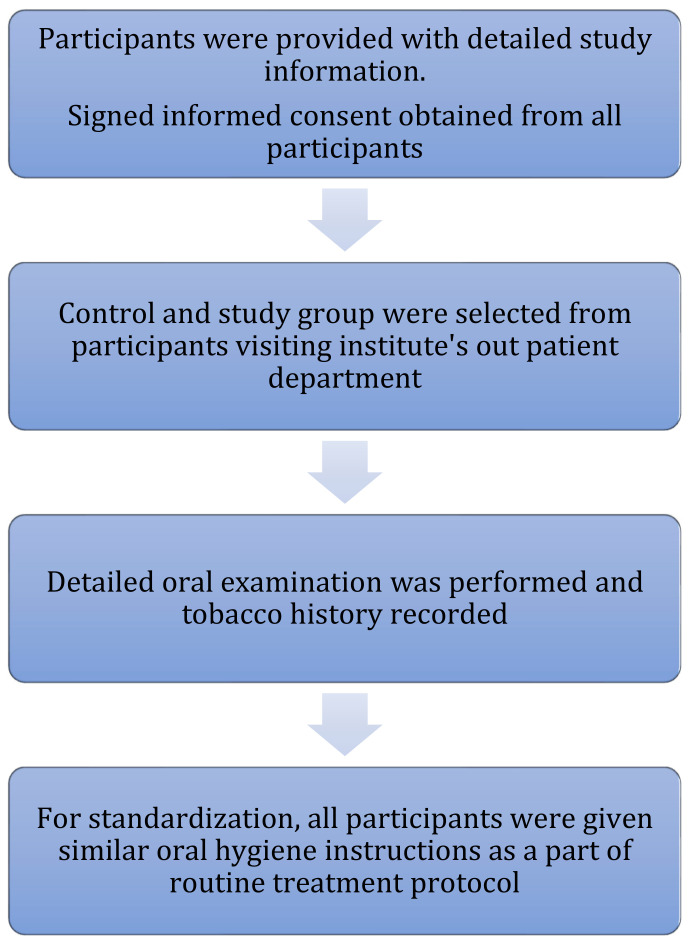
Methodology for clinical examination, Participants information, Consent & uniform instructions.

All parameters of cytomorphometry and *Candida* colonization were evaluated before the application of curcumin in both study and control groups. 

All the tobacco-user participants were counseled to discontinue the habit and were advised to increase their intake of green leafy vegetables [27]. 

Study group participants were given Curenext Gel 50 g tube and were instructed on the application of curcumin gel over lesions three times daily and were called for follow-up after 2 months; meanwhile, in-between follow-up was taken through phone calls asking about consistent application of curcumin gel as per instructions. 

All parameters of cytomorphometry and *Candida* colonization were studied after the application of curcumin in the study group.

### 2.2. Intervention by Curcumin Gel

Curenext Gel tube of 50 g was prescribed (each gram gel containing 10 mg of curcumin extract). It was prescribed to all tobacco users for application three times daily for 2 months at 8 h intervals (3 times per day). Local application of gel on lesions was instructed after meals for a minimum of 20 min. Gel preparation was chosen as it offers better adherence and longer residence time resulting in better exposure to the drug at the site of action.

### 2.3. Cytomorphometric Analysis (Figure 4)

Participants were asked to rinse their mouths thoroughly with water to remove any unwanted debris before mucosal cell collection.Two smears were obtained from each participant by rolling or scrapping uniformly with a cotton swab stick. For the study group, cells were collected from lesion mucosa, and for healthy controls, cells were obtained from buccal mucosa, following the method given by Tolbert et al. [28].Cells from the swab stick were smeared on glass slides to spread collected epithelial cells and were fixed with ethyl alcohol for 10 min and stained with Papanicolaou’s stain (Table 1) (Figure 5).The slides were observed under a light microscope using low magnification (10×) for screening and higher magnification (100×) for scoring (Figure 5, Figure 6, Figure 7 and Figure 8).All smears of both groups were reviewed independently by three examiners and were evaluated according to Papanicolaou from class 0 to class V cytology [29].Cell diameter (CD) and nuclear diameter (ND) were measured along the long axis by the superimposition of calibrated eyepiece graticule, and CD and ND were measured in micrometers. Only in clearly defined cells, and avoiding clumped or distorted cells and nuclei, measurements of individual epithelial cells were made under 100× objective (Figure 6). A hundred cells were measured from each slide and mean values were obtained for CD and ND for each case. Slides were also screened for the presence of genotoxic damage such as micronuclei. Micronuclei were also counted out of 100 intact epithelial cells (Figure 9). All the observations from all three examiners were entered proforma and the average was taken as the final reading. All smears were reviewed independently by all three faculties, which included both professors and assistant professors from the department of oral pathology.

The smears were evaluated and graded following Papanicolaou cytology criteria [29].

Class 0: Lack of adequate sample for assessment;

Class 1: Examination does not reveal any abnormality;

Class 2: Examination does not reveal any abnormality other than inflammatory signs;

Class 3: Smear shows alterations inferring dysplasia (suspect the smear);

Class 4: Examination reveals strong evidence for dysplasia although not confirmatory;

Class 5: Examination reveals malignant features in the smear.

Material insufficient or inadequate for analysis. 

Class 1: Smear appears normal. 

Class 2: Smear appears normal with inflammatory changes. 

Class 3: Smear suspect; dysplastic changes.

Class 4: Strongly shows dysplasia but not conclusive. 

Class 5: Smear shows malignancy. 

**Figure 4 healthcare-10-01507-f004:**
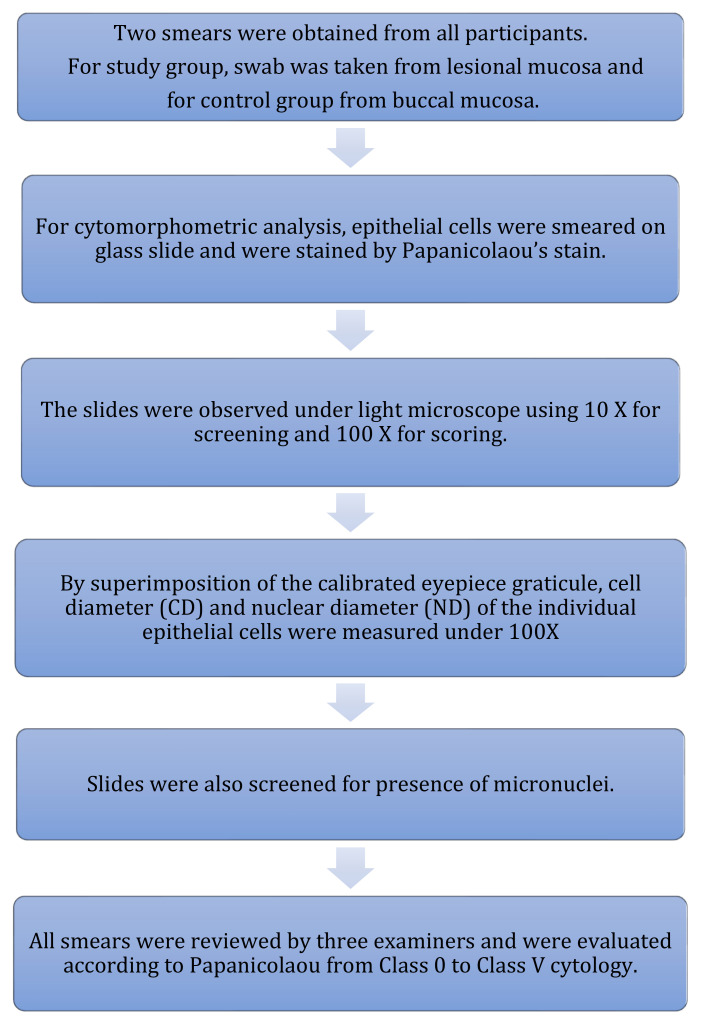
Methodology for cytomorphometric analysis.

The procedure for PAP staining is available in Table 1.

### 2.4. Candida Colonization and Speciation (Figure 10)

Samples for microbiological culture were obtained with a wet sterile cotton swab rolling over the lesion area and then were cultured on Sabouraud’s dextrose agar culture plates [12] (Figure 11). *Candida* colonies on SDA culture plates were counted by using a digital colony counter (Figure 12). Digital colony counting avoided missing colony counting or double counting. These streaked culture plates were incubated at 37 °C for 24 h (Standard procedure for culture was followed). *Candida* detection was also confirmed by Gram staining on light microscopy [13] (Figure 13). Further, they were sub-cultured on the CHROMagar Petri dish and incubated at 37 °C. CHROMagar plates were visualized daily at 24 h, 72 h, and followed for up to 7 days to check for colony growth (Figure 14). *Candida* speciation with different colored creamy colonies was noted. The number of positive cultures of each species was noted in the Performa following *Candida* species color codes [15] (Figure 15 and Figure 16).

**Figure 10 healthcare-10-01507-f010:**
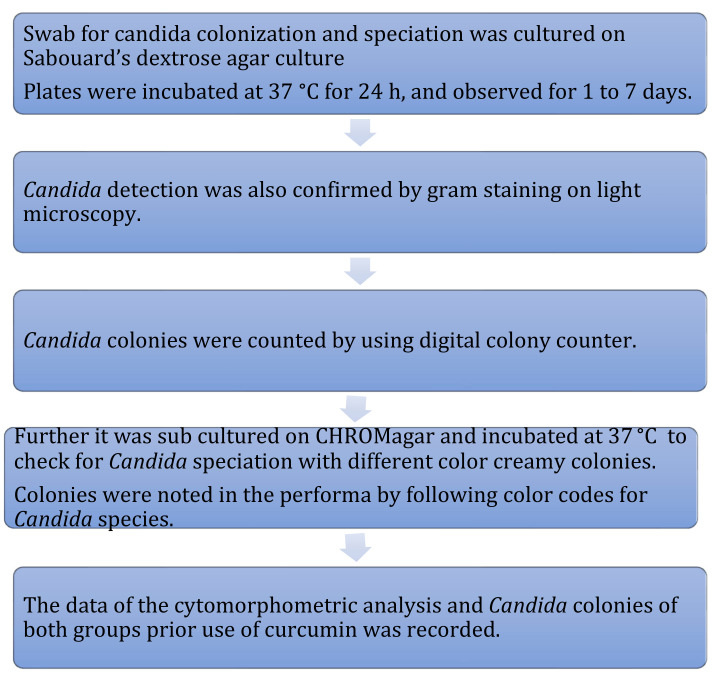
Methodology for *candida* culture and speciation.

*C. albicans*—light green, *C. tropicalis*—blue to purple, *C. krusei*—purple to pink, *C. glabrata*—cream to white, *C. dublinisiasis*—pale green [15].

The data for the colonies of both groups were recorded and evaluated for statistical analysis by using IBM SPSS Version 20.0 (IBM, Armonk, NY, USA). Along with descriptive statistics, significance before and after treatment was calculated by using paired *t* tests. The *p*-value < 0.05 at a 95% confidence interval was considered significant.

**Pre-intervention:** (For evaluation of all parameters of cytomorphometry and *Candida* colonization before application of curcumin in both study and control groups).

### 2.5. Intervention by Curcumin Gel

The participants of the study group were then prescribed Curenext Gel tube of 50 g for application three times daily for 2 months at 8 h intervals (3 times per day) after meals for a minimum of 20 min on lesional mucosa.

Participants were interviewed telephonically regarding the use of curcumin gel over the lesion area.

### 2.6. Post-Intervention

After 2 months of intervention with curcumin gel in the Study group, two swabs were again taken for cytomorphometric analysis and candida colonization and speciation as mentioned above. The same procedure as mentioned above for comparison and evaluation to obtain values for cytomorphometric analysis and candida colonization and speciation before and after the use of curcumin was followed (Figure 17 and Figure 18).

### 2.7. Statistical Analysis 

Sample size was calculated as per number of patients visiting outpatient department of K.M.Shah Dental College and Hospital and considering 2 months’ duration for study.(Figure 19).

Data were statistically analyzed using descriptive statistics.

An independent student’s *t* test was employed to compare the study groups. The comparative groups were assessed concerning parameters including the dimensions of the cell and the nucleus. The ratio of the nucleus to the cytoplasm was also analysed, along with candidal prevalence and species specification.

Additionally, pre- and post-intervention cytomorphometric parameters and candida colonization were compared by using a dependent *t* test.

Data analysis was by using SPSS Version 22.0 (IBM, Armonk, NY, USA). Statistical significance was set at *p* < 0.05.

## 3. Results

In both the study and control groups there was no significant difference in the mean age of male and female participants (Table 2).

Out of the 60 participants in the study group, 31 consumed smokeless tobacco, 9 consumed smoking tobacco, and 20 consumed smokeless and smoking tobacco (Table 3).

In the study group, Leukoplakia was predominant among the various lesions observed in patients with tobacco abuse; 35% of patients were clinically diagnosed with leukoplakia followed by 30% with tobacco pouch keratosis, 20% with oral submucous fibrosis (OSMF), and 15% with smoker’s palate (Table 4).

The mean nuclear diameter was increased in the study group (9.95 ± 0.86) as compared to the control group (8.27 ± 0.66), and the difference was statistically significant (*p* = 0.001).

The mean nuclear diameter was reduced after intervention in the study group and the difference was statistically significant (*p* = 0.098).

The mean cell diameter was increased in the control (66.95 ± 1.76) group as compared to the study group (63.69 ± 1.64), and the difference was statistically significant (*p* = 0.001). The mean cell diameter before and after the intervention in the study group did not show a statistically significant difference.

The mean CD:ND ratio is higher in the control group as compared to the study group and is statistically significant (*p* = 0.001). The mean CD:ND ratio was increased post-intervention in the study group and was statistically significant (*p* = 0.073).

The mean micronuclei number was increased in the study group (1.86 ± 1.04) as compared to the control group (0.37 ± 0.49) and the difference was statistically significant (*p* = 0.001). The mean micronuclei were found to be reduced post-intervention in the study group and the difference was statistically highly significant (*p* = 0.000).

The class of cytology (indicating epithelial dysplasia) was found to be higher in the study group as compared to the control group and the difference was statistically significant (*p* = 0.001).

Pre- and post-intervention differences in the class of cytology were found to be statistically significant in the study group (*p* = 0.001).

*Candida albicans* was predominant in the control group (44; 73.33%) when compared with the study group before intervention, and the difference was statistically highly significant (*p* = 0.0001), whereas *Candida tropicalis* was found to be predominant in the study group (37; 61.67%) as compared to control and the difference was statistically highly significant (*p* = 0.0001).

Pre-intervention differences between the number of participants found with *Candida glabrata* and *Candida krusei* in both the groups were not statistically significant.

Before the use of curcumin, *Candida tropicalis* was found predominant in the study group (n = 37, 61.67%) followed by *Candida albicans* (n = 12, 20.00%), whereas post-intervention (after application of curcumin gel for 2 months), *Candida albicans* (n = 42, 70.00%) was found predominant, followed by non-Albicans candida tropicalis (n = 16, 26.67%). The difference between *Candida albicans* pre- and post-intervention in the study group was statistically highly significant (*p* = 0.00004).

The pre- and post-intervention difference in *Candida tropicalis* was highly significant (*p* = 0.0039).

The occurrence of other Species of *Candida* in the Study group such as *Candida glabrata* and *Candida krusei* had decreased after the use of Curcumin with a significant difference in *Candida glabrata* (*p* = 0.0114), but pre and post-intervention difference for *Candida krusei* were statistically not significant.

It was observed that the mean number of *Candida* colonies (colony forming units, CFU) was greater in the study group (10.57 ± 2.68) than in the control group (3.88 ± 1.43), and the difference was statistically highly significant (*p* = 0.001) (Table 5).

In the present study, 21 participants had leukoplakia, oral submucous fibrosis was seen in 12 participants, and smoker’s palate was seen in 9 individuals, whereas tobacco pouch keratosis was diagnosed in 18 participants based on clinical assessment and history of habit.Distribution of cytomorphometry parameters such as CD, ND, and CD:ND ratio was altered before and after intervention, but the difference was not statistically significant in all three lesions, except for ND (*p* = 0.022) and CD:ND ratio (*p* = 0.021) with the significant difference in tobacco pouch keratosis.The micronuclei frequency was altered from pre-intervention to post-intervention with a significant reduction in all four groups.


**Prior to intervention:**
4.It was also observed that type of *candida* on CHROMagar in the patients of leukoplakia predominantly inhabited *C. tropicalis* (13), followed by *C. albicans* (06), *C. glabrata* (1), and *C. krusei* (1).5.Twelve patients were diagnosed with OSMF out of which eight cases were of *C. tropicalis* followed by *C. albicans* (2) and *C. glabrata* (1) and *C. krusei* (1).6.Nine patients were diagnosed with smoker’s palate out of which six participants inhabited *C. tropicalis,* followed by *C. glabrata* (3 cases).7.Eighteen patients were diagnosed with tobacco pouch keratosis out of which ten participants had *C. tropicalis,* followed by *C. albicans* (4) and *C. glabrata* (4).



**Post Intervention:**
8.It was found that in leucoplakia there was a shift in the type of *candida* species, *Candida albicans* (11) was predominant, followed by *Candida tropicalis* (8) and then *C. glabrata* (1) and *C.krusei* (1), and the difference was statistically significant in *C. albicans* and *C. tropicalis*.9.In oral submucous fibrosis, there was the predominance of *Candida albicans* (09), followed by *Candida tropicalis* (03), and pre- and post-intervention differences for both types of candida were statistically significant.10.For smoker’s palate, there was a predominance of *Candida albicans* (07), followed by *Candida tropicalis* (02), which were statistically significant (*p* = 0.001).11.In tobacco pouch keratosis, there was a predominance of *Candida albicans* (15), followed by *Candida tropicalis* (03), and the difference was statistically significant (*p* = 0.001).


Thus, *Candida* is predominantly associated with leukoplakia, followed by tobacco pouch keratosis, oral submucous fibrosis, and smokers’ palate

There was a significant reduction in the total number of candida colonies in all four lesions of tobacco users on post-intervention of curcumin when compared to pre-intervention candidal colonies (Table 6).

In the control group, all the participants belonged to Class I cytology, and predominantly *C. albicans* (44) was detected on CHROMagar, followed by *C. tropicalis* (11), followed by *C. glabrata* (4) and *C. krusei* (1).

In the study group prior to intervention:

1.There were two patients of class I cytology and the type of candida on CHROMagar was found to be *C. tropicalis*.2.Thirty-one patients of Class II cytology predominantly had *C. tropicalis* (17), followed by *C. albicans* (8), *C. glabrata* (4), and *C. krusei* (2).3.Twenty-seven patients had Class III cytology, and the species identified was predominantly *C. tropicalis* (18), followed by *C. glabrata* (5), and *C. albicans* (4), and no species of *C. krusei* was observed.

In study group post-intervention:There were 13 cases of class I cytology on CHROMagar found with *C. albicans* (8), followed by *C. tropicalis* (5).There were 25 cases of Class II cytology, found predominantly with *C. albicans* (19), followed by *C. tropicalis* (5) and *C. krusei* (1).There were 22 cases of Class III cytology, found predominantly with *C. albicans* (15), followed by *C. tropicalis* (6) and *C. glabrata* (1).

Thus, it was observed that the participants of the study group post-intervention predominantly shifted from *C. tropicalis* to *C. albicans* and from class Cytology II and III to Class I. (The same results were found in the control group) (Table 7).

Among the 60 participants of the study group, 31 were smokeless tobacco chewers, 20 had both smokeless and smoking habits, whereas 9 participants were smokers.

In our study, *candida tropicalis* was found more in the smokeless tobacco (18) consuming group followed by *candida albicans* (8), *candida glabrata* (4), and *candida krusei* (1). After intervention, *candida albicans* (24) was more prevalent, followed by *candida tropicalis* (7).

In the smoking tobacco consuming group, *candida tropicalis* was found prevalent (5), followed by *candida glabrata* (3) and *candida albicans* (1). After intervention, *candida albicans* (7) was more prevalent, followed by *candida tropicalis* (2).

In the study group, participants consuming both the smokeless and smoking type of tobacco had *candida tropicalis* prevalent (14), followed by *candida albicans* (3), *candida glabrata* (2), and *candida krusei* (1). After intervention, *candida albicans* (11) was more prevalent, followed by *candida tropicalis* (7), *candida glabrata* (1), and *candida krusei* (1).

In all three study groups, we found detection of *candida tropicalis* commonly as compared to other species of *candida*, and in all the study groups with habits, irrespective of the type of tobacco habit, post-intervention, there was shift of *candida tropicalis* to *candida albicans* (Table 8).

## 4. Discussion

“Good oral health is the gateway to good systemic health”. Thus, oral health is a very important component of general health. Tobacco and related substances have a significant impact on the overall health of individuals and the community. Tobacco has a strong socioeconomic impact causing oral lesions leading to functional limitation, pain, and discomfort, leading to pre-cancer and cancer. Tobacco habits are detrimental to human health and toxic chemicals of tobacco, including nicotine, invade multiple systems resulting in physical, psychological, and social disability, affecting the quality of life of people adversely [30].

As per our results in the habit group, there was a predominance of male participants involving 60% of males and 40% of females (Table 2) and it was following the study of Smita Asthana et al. who revealed up to one-fourth of cancers among males and lower than one-fifth among females were tobacco-related [31].

It was seen that the frequency, distribution, and type of tobacco consumption have an effect on the development of pre-malignant lesions. Our study group has more participants with smokeless tobacco consumption habits (51.6%) than smoking (15%) and both smokeless and smoking (33.4%) [32,33] (Table 3 and Table 4).

Very early detection of potentially malignant oral lesions is most important for better prognosis and survival of patients with oral cancer. Along with a visual examination of lesions, exfoliative cytology gives better insight into cellular and nuclear details. Malignant transformation induced by tobacco use is reflected in the nucleus of the affected cells [34].

Any cell exposed to the tobacco products would have been imparted with potential malignant changes which could be elicited in exfoliative cytology.

Our study evaluated the cytomorphometric changes in the buccal mucosa of tobacco users and found an increased incidence of micronuclei in exfoliated oral mucosal cells, a significant reduction of mean cell diameter, and an enlarged mean nuclear diameter and N:C ratio seen in tobacco abusers when compared with the control group (Table 5 and Table 6). This finding was congruent with a study by Hastak et al. [35] who studied micronuclei in OSMF patients. The CD and ND findings of our study were in congruence with Srilatha et al. [36], Shetty et al. [37], Mahadoon et al. [8], Kokila S. et al. [2], Mollaoglu et al. [38], Cowpe et al. [39], Ogden and Cowpe [40], and Ramesh [41], who found that there is an increase in the ratio of nucleus to cytoplasm in tobacco users when compared to the control, while there is an attenuation of the diameter of the cell and the nucleus. In our study, almost all tobacco users’ lesions showed Class II and Class III cytology suggesting epithelial atypia in tobacco users as compared to normal healthy participants. Ramesh et al. [42]. found that dysplasia and oral cancer cases had an augmented diameter of the nucleus compared to normal cells of the oral mucosa. Rashmi et al. [10] also revealed a statistically significant difference between mean nuclear diameter values for the normal control group and pre-cancer and cancer patients.

Incidence of micronuclei has been analyzed by various studies in normal patients, oral pre-malignancy, and OSCC [10,42]. In the present study, micronuclei in oral exfoliated cells in the control group had a mean micronucleus frequency of 0.37 ± 0.49, and these levels are quite similar to those reported by Palve and Dave et al. [43,44]. The overall level of micronuclei was increased in control pre-cancer patients and cancer patients in a study by Saran et al. [45]. Constant contact of tobacco and tobacco products with oral mucosa induces a local rise in temperature and inflammation for longer periods, resulting in delayed cell division, which, in turn, increases the nuclear diameter as there may be an increase in nuclear content during replication. In such cells with increased activity, the ability of cells to form cytoplasm decreases. The increased nuclear volume and decreased cytoplasmic volume possibly represent the significant changes that occur in the cells, which are more accurately identified at a morphological level [46]. Another mechanism suggested that tobacco consumption releases various by-products such as nitrosamine and nicotine, which can influence the cellular morphometry [34]; also in our study, the class of cytology inference was Class I, indicating normal cellular and nuclear features in the control group, whereas in the case of the study group, the pre-intervention class of cytology was Class II and III in the majority of cases, suggesting early dysplastic changes due to tobacco habits. Post-curcumin assessment of cytomorphometry revealed Class I cytology in more cases as compared to Class II and III pre-intervention cytomorphometry analysis, suggesting an effect of curcumin as stated in various studies [35,47,48]. In our study, post-intervention cytomorphometric evaluation of curcumin showed changes (Table 6) may occur as curcumin mediates its anti-pre-cancer activities by increasing levels of vitamins C and E and preventing lipid peroxidation and DNA damage [42]. Additionally, the apoptosis induction effect of curcumin and its analogs on cancer cell lines have shown the role of curcumin as a potent anti-carcinogenic polyphenol [38,40,41].

Curcumin possibly activates mitochondrial enzymes that lead to the production of reactive oxygen species (ROS) through interaction with thioredoxin reductase at higher doses, whereas, at the lower dose of curcumin, it quenches ROS production and acts as an antioxidant by inhibiting free radicals [19]. Additionally, curcumin acts as an inhibitor of the TNF-α-induced NF-κB signaling pathway. Curcumin also acts on oral mucositis by reducing the secretion of inflammatory chemokines and reducing the toxic effects of bacteria by its local application [49].

Curcumin’s immunomodulatory action is largely mediated by the attenuation of COX2, a key enzyme involved in inflammation and cell renewal [50].

Thomas et al. [51]. inferred that application of a gel containing curcuminoid (1%) in the form of a topical application did not have a significant effect compared to the application of triamcinolone acetonide (0.1%) in attenuating the symptoms resulting from the lesions. On the contrary, Vibha et al. [52] inferred that the application of curcuminoid in the form of an ointment was sufficient to relieve the symptoms.

The collection of specimens for *Candida* inoculation and speciation in our study on CHROMagar plates by using sterile swabs was similar to Baradkar et al. [15,53].

In our study, it was observed that *Candida* colonies were increased in samples of the study group compared to the control group, and colony growth appeared within a shorter duration in the study group than in the control group (Table 6). Higher *Candida* colonization, using the swab technique, in tobacco users was in agreement with studies reported by Keten et al. [54]. Our finding of higher *Candida* colonization in the study group was in agreement with the study reported by Alaizari et al. [55], who assessed colonization of *Candida* in both non-smokers and smokers. Higher oral *Candidal* colonization in tobacco abusers as compared to non-tobacco users may be because of altered polymorphonuclear leukocyte activity and functions in turn leading to the proliferation of the *Candida* species [55]. Additionally, the nicotine present in saliva constitutes a suitable vehicle for further Candida proliferation and growth. Thus, tobacco leads to increased *Candida* concentration rather than true *Candidal* association.

All these tobacco-induced alterations may be because of the vicious cycle of alterations of the mucous membrane, further amplification of pro-Candida substances added by altered acidic pH of saliva seen in tobacco users and hence increased Candida [56] growth associated with already compromised defense mechanisms.

A statistically significant reduction in *Candida* colonies after intervention with curcumin was seen in our study and may be due to the stronger antifungal activity of curcumin. The role of the anti-*candida* activity of curcumin, along with the maintenance of oral hygiene and the discontinuation of habit, cannot be denied.

Candida albicans is the most commonly detected species of candida in humans. The prevalence of oral candida in systemically healthy individuals varies from 17% to 75% [17,54].

In the present study, *C.albicans* was the predominantly identified species (Table 7), and our results were compatible with those of Rashmi et al. [13] and Krogh et al. [57]. 

Rashmi et al. [13] found candida albicans as the predominant species identified. They also found a highly significant association of candida more often in cancer patients than in pre-cancers in both cytopathology and histopathology when identified by using calcofluor white along with routine diagnostic methods. Krogh et al. [57] found the association of yeasts in 82% of leukoplakias and 37% of lichen planus patients, and they also found candida Albicans as the dominant species identified in both lesions.

Other species detected by Krogh et al. [57] were *Candida tropicalis* and *Candida pintolopesii,* followed by *Torulopsis glabrata* and *Saccharomyces cerevisiae*.

The presence of *C. albicans* as the predominant species in the healthy control group (Table 8) in our study was congruent with the study of Keten et al. [54]. *Candida* exacerbation due to tobacco use may be because of increased salivary glucose levels, especially as tobacco substances [17].

Before the use of curcumin, in a study group of tobacco abusers, *Candida tropicalis* was identified in more cases (61.67%), followed by *Candida albicans* (20.00%), whereas, after the use of curcumin, *Candida albicans* (70.00%) was seen followed by *Candida tropicalis* (26.67%). In the tobacco habit group, the presence of increased non-*Candida albicans* species in our study was similar to a study by Sonia Silva et al. [58]. The presence of *Candida tropicalis* has been reported by Keten et al. [54] and Krogh et al. [57]. The *C. tropicalis* strain has been reported similarly in studies by Keten et al. [54]. The occurrence of *Candida tropicalis* is a common finding in tobacco users, whereas *Candida albicans* is found in non-tobacco users.

In our study pre-intervention, *Candida tropicalis* was found, whereas post-intervention there was shift of *Candida tropicalis* to *Candida albicans* (which is also commensal of the oral cavity) in the habit group irrespective of the type of habit, i.e., smokeless tobacco or the smoking form of tobacco.

In our study, there was a reduction in the number of candida colonies after the intervention with curcumin, depicting antifungal property of curcumin. It has been seen that curcumin has antifungal effect on *candida tropicalis*, *candida albicans*, *candida glabarata,* and *candida krusei* [59,60].

In addition, curcumin’s function is mediated through the leakage of components within the cell via a compromised membrane. Lee et al. [61] found that cellular homeostasis is essential to life, and it is achieved by controlling the permeability of the cytoplasmic membrane to ions and solutes. They predicted the role of the potassium ion gradient in regulating pH and cell structure and as an important determinant of its growth. The study by Lee et al. suggested that curcumin exerts antifungal activity via inducing disruption of the fungal plasma membrane. They found that loss of cytoplasmic potassium leads to cell death in fungi [61].

The extrusion of H+ was inhibited by a curcumin-mediated increase in the acidity within the cells which induced apoptosis of *Candida* [17]. In addition, curcumin inhibited hyphae by targeting the global suppressor thymidine uptake 1 (TUP1) [17]. Turmeric (methanol extract) attenuation of *Candida* provides further evidence of the same [17].

**The limitations of our study** include the short duration of the intervention as a part of the STS-ICMR project, the discontinuation of the habit along with oral hygiene maintenance instructions followed by curcumin administration and that the changes in cytomorphometry and *Candida* speciation due to curcumin were in association with the discontinuation of habit counseling and thus cannot be considered as a sole effect of curcumin and needs to be further analyzed over a longer duration and with frequent follow-ups.

The other conventional methods of candida culture, other than SDA, such as germ tube and chymadospore formation, were time consuming so were not compared with CHROMagar. We used SDA, which is considered the gold standard and whose sensitivity is considered equal to conventional methods [12].

## 5. Conclusions

Screening of oral lesions by using simple, non-invasive, rapid PAP stain and cytomorphometry evaluation is very important to create awareness and educate tobacco users with counseling for the discontinuation of habits and should be done at regular intervals to avoid pre-cancer and cancer. Curcumin is found to reduce *Candidal* colonization along with discontinuation of habit and maintenance of oral health in tobacco and related substance users, which is very important, especially in the era of microbial resistance, supporting its antifungal activity as mentioned in various in vitro and clinical studies. To know its exact mechanism, a follow-up study with a longer duration is required. Additionally, CHROMagar was found to be convenient for rapid *Candida* speciation. Further, more studies are necessary to understand the shift from *c. tropicalis* to *c. albicans* in tobacco users after the use of curcumin [12].

## Figures and Tables

**Figure 1 healthcare-10-01507-f001:**
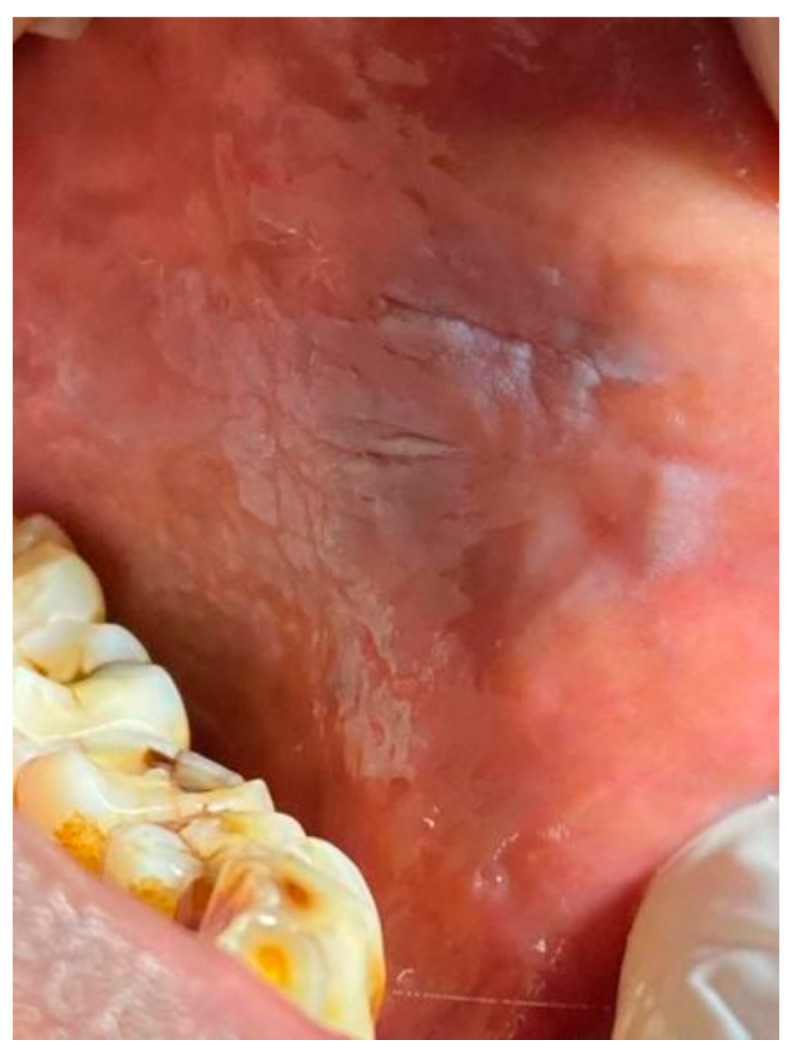
Tobacco user’s lesion on left buccal mucosa leukoplakia.

**Figure 2 healthcare-10-01507-f002:**
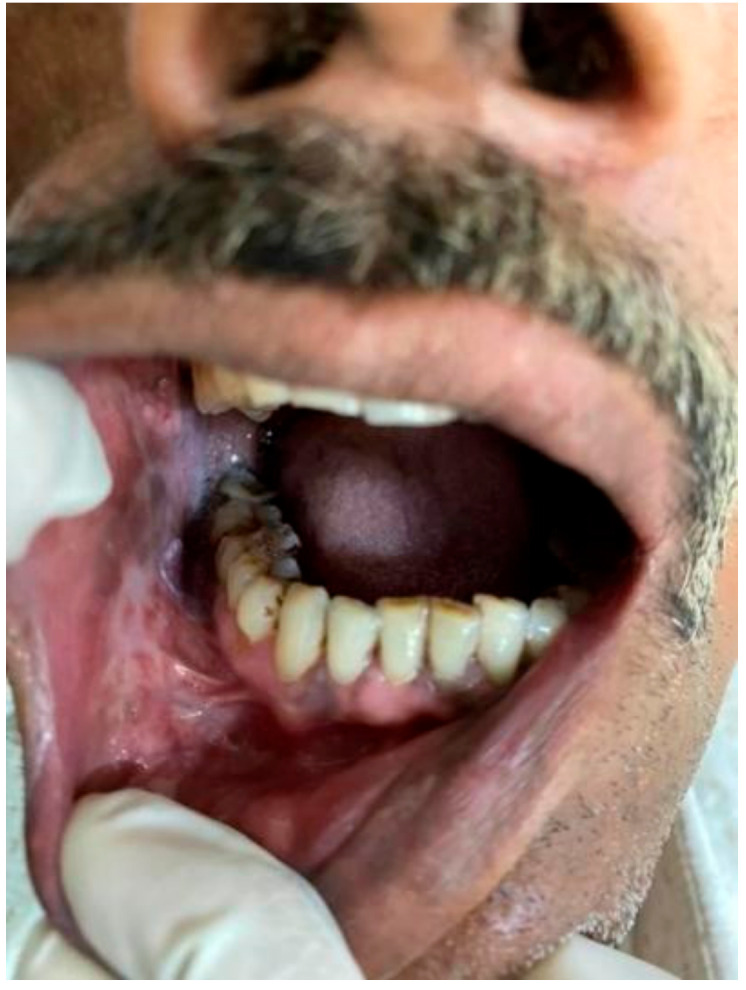
Tobacco user’s lesion on right buccal mucosa along with oral submucous fibrosis.

**Figure 5 healthcare-10-01507-f005:**
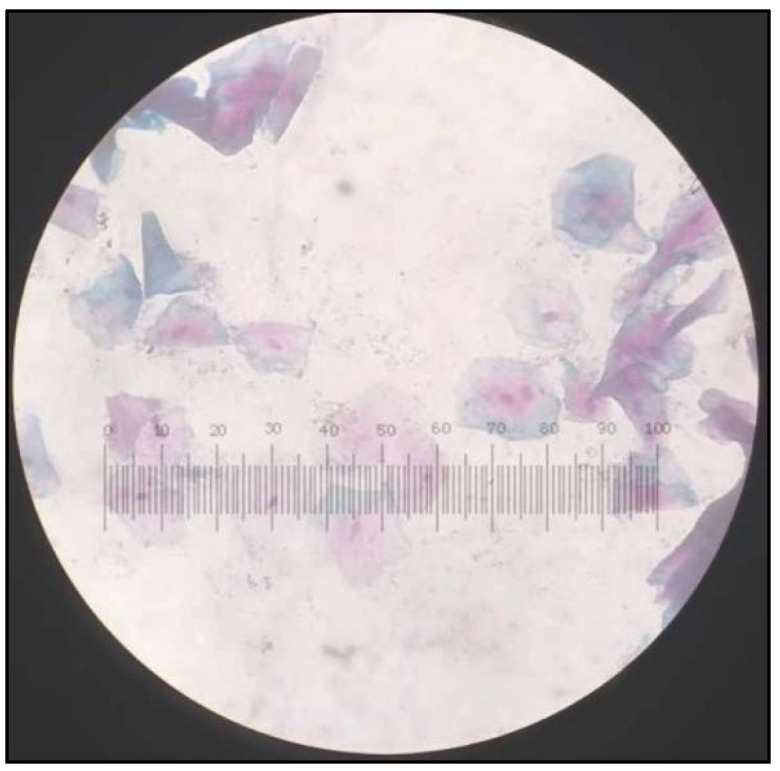
Papanicolaou Stained (PAP) smear showing cells from control group with Class I cytology.

**Figure 6 healthcare-10-01507-f006:**
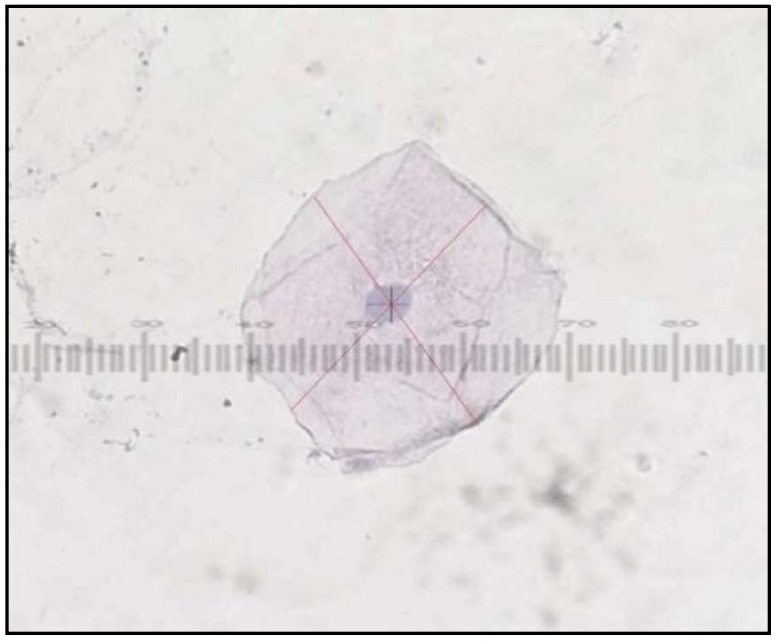
PAP smear showing cells from control group with Class I cytology cell diameter and nuclear diameter measurements.

**Figure 7 healthcare-10-01507-f007:**
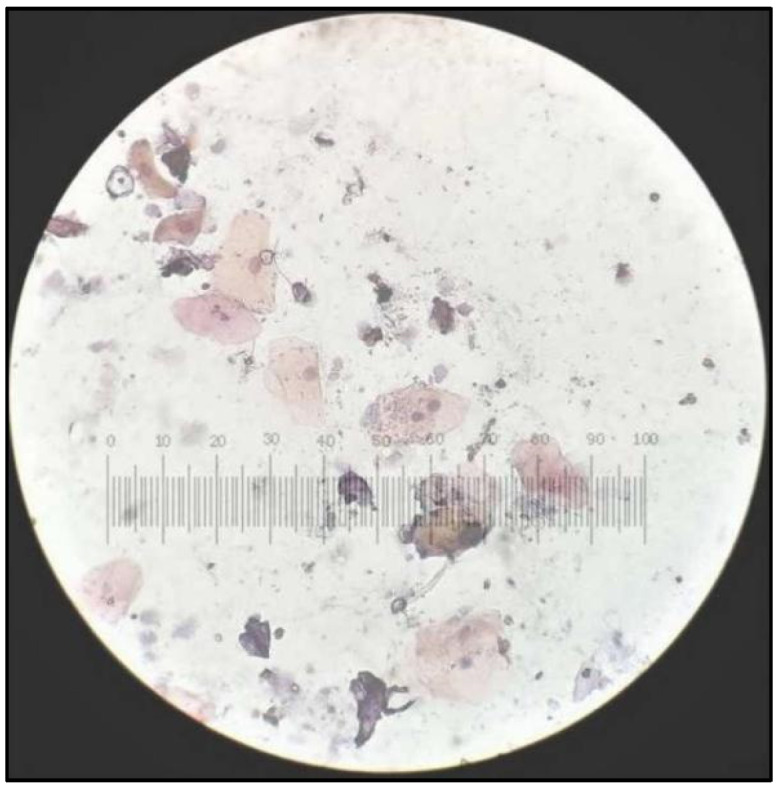
PAP smear showing cells from tobacco-user group with Class I cytology.

**Figure 8 healthcare-10-01507-f008:**
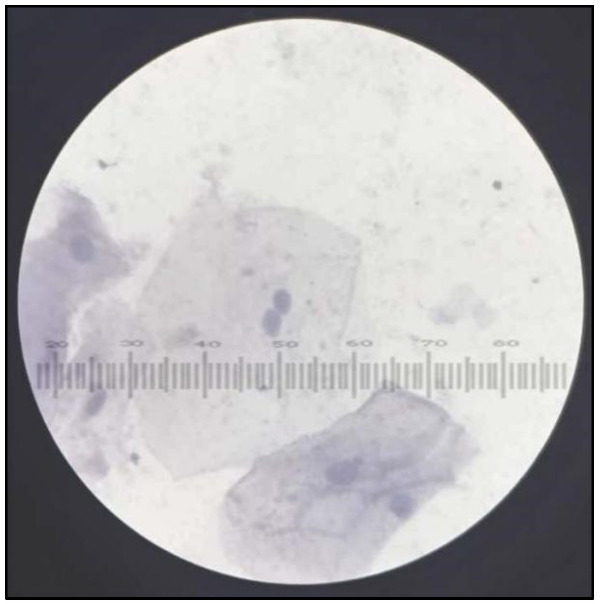
PAP smear showing cells from tobacco-user group with Class II cytology.

**Figure 9 healthcare-10-01507-f009:**
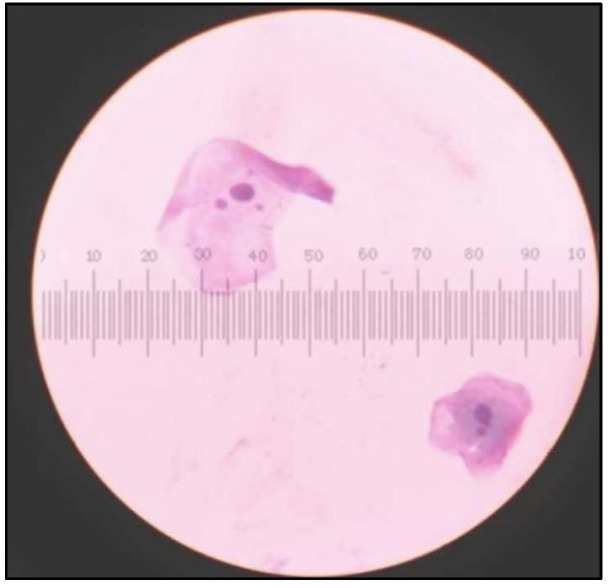
Cells showing micronuclei in cells from tobacco abusers.

**Figure 11 healthcare-10-01507-f011:**
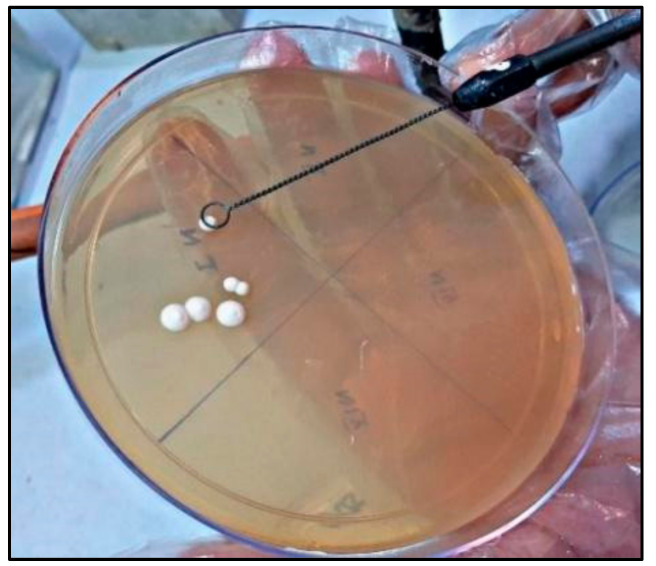
*Candida* colonies on Sabouard’s dextrose agar culture plates.

**Figure 12 healthcare-10-01507-f012:**
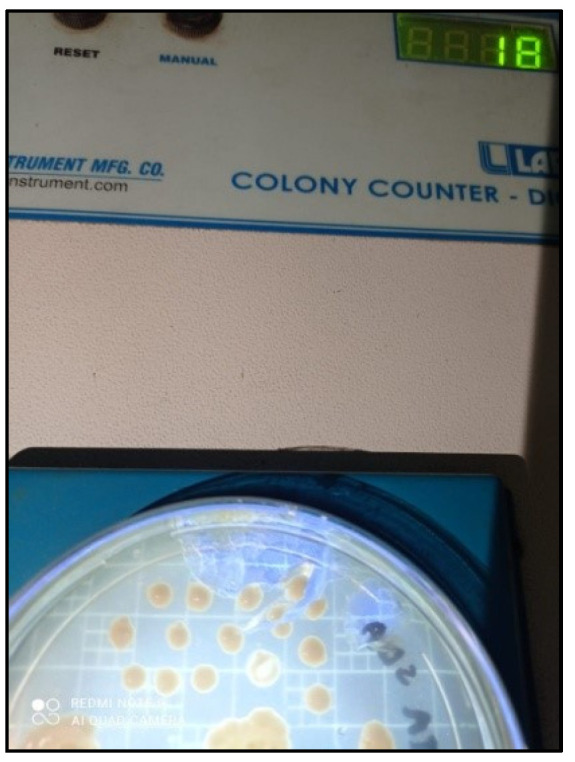
*Candida* colony count on digital colony counter.

**Figure 13 healthcare-10-01507-f013:**
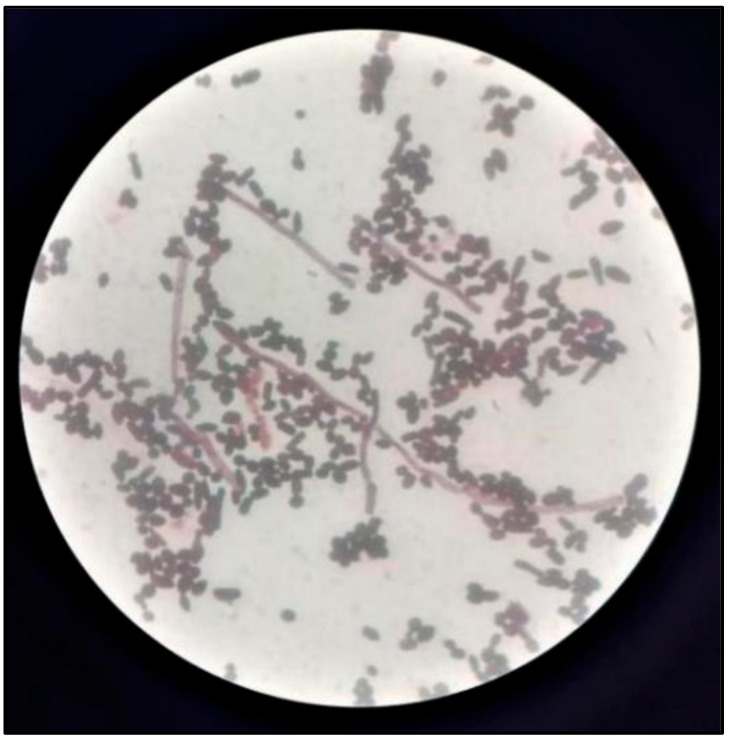
Gram staining for detection of *Candida*.

**Figure 14 healthcare-10-01507-f014:**
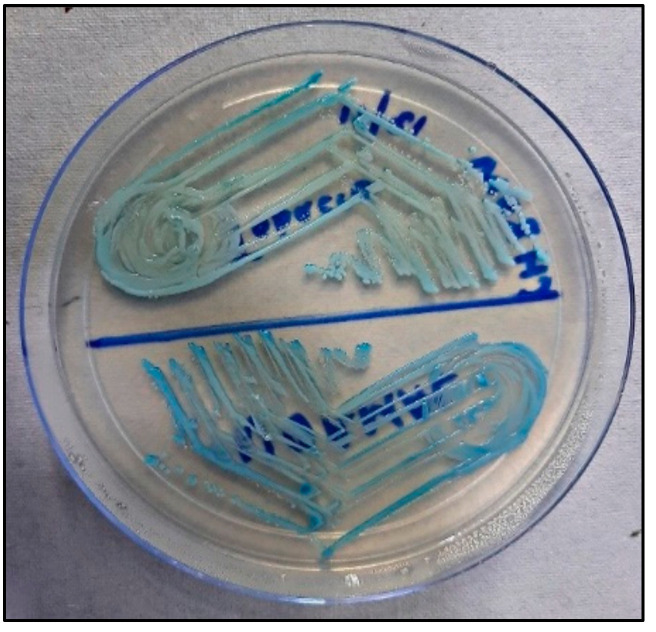
*Candida albicans* and *Candida tropicalis* on CHROMagar.

**Figure 15 healthcare-10-01507-f015:**
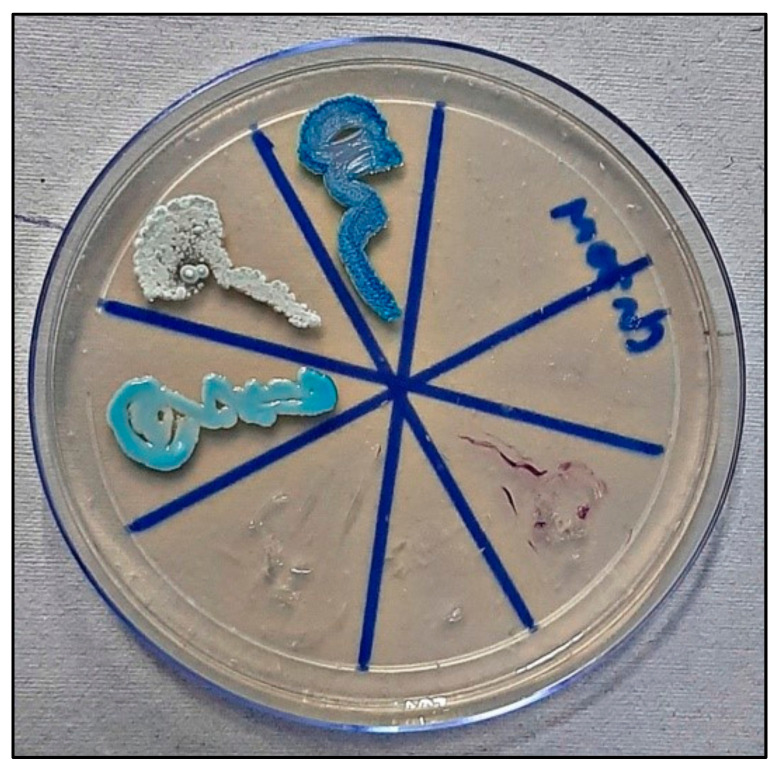
*Candida* colonies on CHROMagar in control group.

**Figure 16 healthcare-10-01507-f016:**
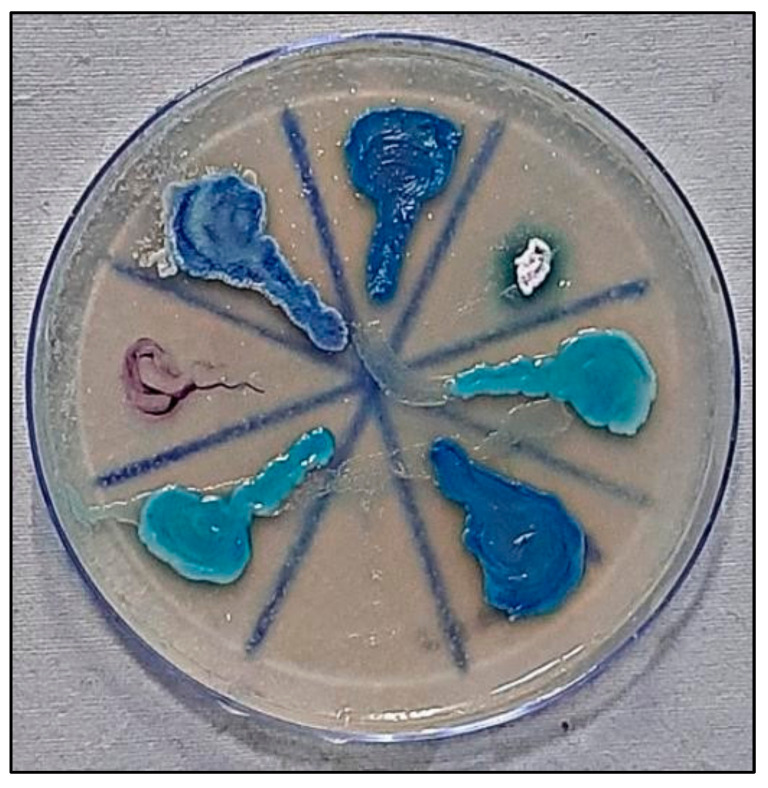
*Candida* colonies on CHROMagar in tobacco-user group.

**Figure 17 healthcare-10-01507-f017:**
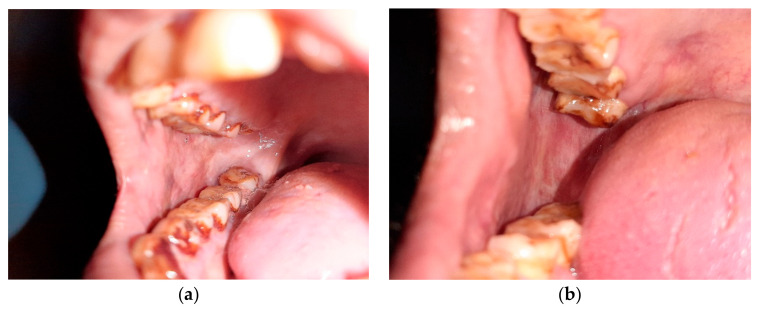
Tobacco Chewer’s Keratosis Prior (**a**) and After (**b**) the use of Curcumin.

**Figure 18 healthcare-10-01507-f018:**
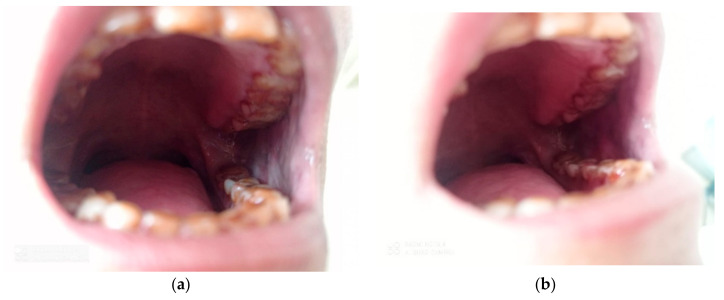
Hyperkeratotic Lesion on Left Buccal Mucosa Prior (**a**) and After (**b**) Intervention with Curcumin.

**Figure 19 healthcare-10-01507-f019:**
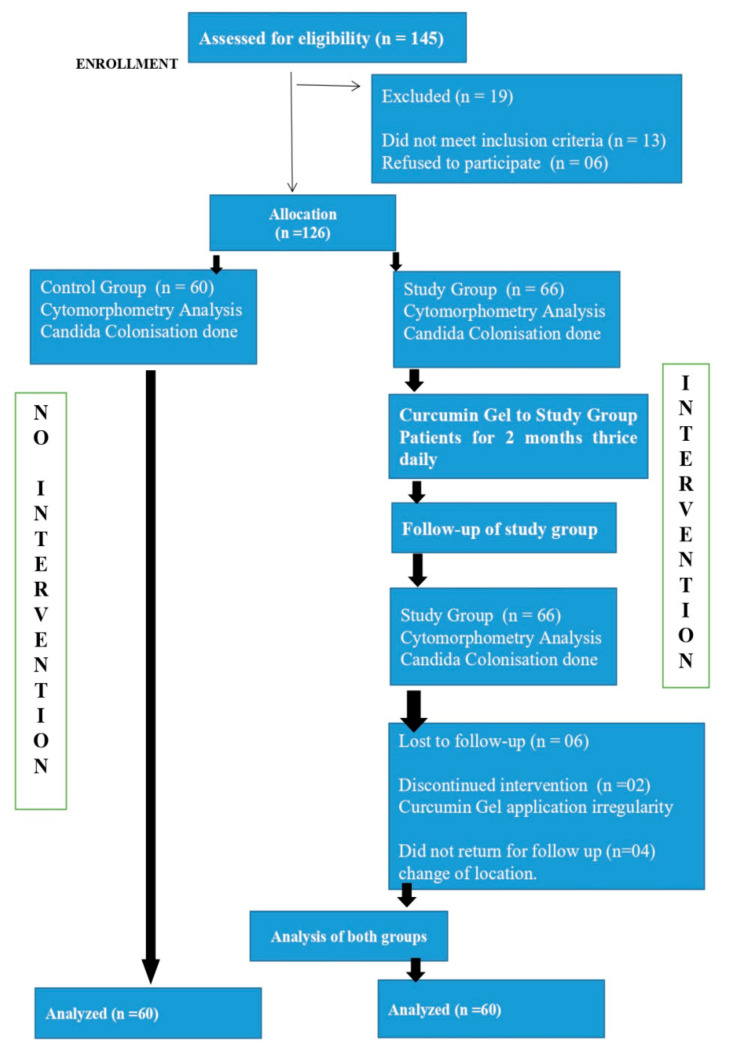
CONSORT Flow Chart for study.

**Table 1 healthcare-10-01507-t001:** Procedure for pap staining.

Agent	Staining Time
Running water	1 min
Harris hematoxylin	6 min
Running water	1 min
Aqueous HCL solution	
6 dips	
Running water	1 min
Lithium carbonate	
2 min	
50% ethanol	10 dips
95% ethanol	6 to 8 dips
95% ethanol	6 to 8 dips
Orange G-6	1.5 min
95% ethanol	Rinse gently
95% ethanol	Rinse gently
Acid–Eosin 50	1.5 min
95% ethanol	Rinse gently
95% ethanol	Rinse gently
100% ethanol	6 to 8 dips
100% ethanol	6 to 8 dips
Xylene	6 to 8 dips
Xylene	6 to 8 dips

**Table 2 healthcare-10-01507-t002:** Age and gender distribution in study and control groups.

Group	Gender	N	Mean	SD	*p* Value
Study Group	Male	36	34.31	5.569	0.064
	Female	24	31.33	6.190	
Control Group	Male	31	29.61	5.475	0.619
	Female	29	28.33	5.092	

**Table 3 healthcare-10-01507-t003:** Distribution of type of tobacco in study group.

Type of Tobacco Habit in Study Group	No. of Patients
Smokeless Tobacco	31 (51.6%)
Smoking Tobacco	09(15%)
Smoking and Smokeless Tobacco	20(33.4%)
Total	60

**Table 4 healthcare-10-01507-t004:** Type of lesion based on clinical assessment.

Type of Lesion Based on Clinical Assessment	Number of Participants
Leukoplakia	21 (35%)
OSMF	12 (20%)
Smokers Palate	9 (15%)
Tobacco Pouch Keratosis	18 (30%)
Control Group	60 (100%)

**Table 5 healthcare-10-01507-t005:** Comparison of cytomorphometry and microbial parameters among both study and control groups as well as pre- and post-intervention groups.

Cytomorphometric Analysis (Pre-Intervention)	Study Group		Control Group		*p*-Value	Pre-Intervention		Post-Intervention-		*p*-Value
	Mean µm	SD	Mean µm	SD		Mean	SD	Mean	SD	
Nuclear Diameter	9.95	0.86	8.27	0.66	0.000	9.95	0.86	9.70	0.78	0.098
Cell Diameter	63.69	1.64	65.95	1.76	0.000	63.69	1.64	63.91	1.509	0.444
CD:ND Ratio	6.44	0.60	8.01	0.62	0.000	6.44	0.60	6.63	0.553	0.073
Micronuclei	1.86	1.04	0.37	0.49	0.000	1.86	1.04	0.88	0.555	0.000
*C. albicans* (*N*)	44 (73.33%)		12 (20.00%)		0.000	12 (20.00%)		42 (70.00%)		0.000
*C. tropicalis* (*N*)	11 (18.33%)		37 (61.67%)		0.000	37 (61.67%)		16 (26.67%)		0.003
*C. glabrata* (*N*)	4 (6.67%)		9 (15.00%)		0.165	9 (15.00%)		1 (1.67%)		0.011
*C. krusei* (*N*)	1 (1.67%)		2 (3.33%)		0.563	2 (3.33%)		1 (1.67%)		0.563
Number of Microbial Colonies	10.57	2.68	3.88	1.43	0.001	10.57	2.68	5.02	2.38	0.000
Colony Appearance Duration (Number of Days)	1.15	0.36	2.20	0.73	0.001	1.15	0.36	2.02	0.70	0.000

**Table 6 healthcare-10-01507-t006:** Distribution of cytomorphometry and microbial parameters in tobacco lesions as per clinical assessment.

	Leukoplakia			OSMF			Smokers Palate			Tobacco Pouch Keratosis		
	Pre	Post	*p* Value	Pre	Post	*p* Value	Pre	Post	*p* Value	Pre	Post	*p* Value
**CD**	63.89 ± 1.63	64.01 ± 1.50	0.805	63.48 ± 1.63	63.74 ± 1.50	0.688	63.77 ± 1.63	64.03 ± 1.51	0.730	63.58 ± 1.60	63.87 ± 1.49	0.577
**ND**	9.97 ± 0.86	9.87 ± 0.80	0.698	9.86 ± 0.85	9.79 ± 0.77	0.654	9.90 ± 0.84	9.85 ± 0.79	0.898	10.03 ± 0.87	9.372 ± 0.78	0.022
**CD:ND**	6.43 ± 0.60	6.543 ± 0.56	0.542	6.48 ± 0.59	6.54 ± 0.55	0.799	6.54 ± 0.59	6.55 ± 0.56	0.971	6.38 ± 0.61	6.851 ± 0.56	0.021
**Micronuclei**	1.95 ± 1.06	0.71 ± 0.57	0.0001	1.41 ± 1.03	0.83 ± 0.55	0.099	2.33 ± 1.04	1.11 ± 0.57	0.007	1.83 ± 1.05	1 ± 0.56	0.005
** *C. albicans* **	06 ± 0.69	11 ± 0.63	0.000	02 ± 0.57	09 ± 0.61	0.000	-	07 ± 0.61	-	04 ± 0.75	15 ± 0.61	0.000
** *C. tropicalis* **	13 ± 0.65	08 ± 0.62	0.000	08 ± 0.70	03 ± 0.48	0.000	06 ± 0.59	02 ± 0.50	0.000	10 ± 0.65	03 ± 0.63	0.000
** *C. glabrata* **	01	01	-	01	-	-	03 ± 0.67	-	-	04 ± 0.71	-	-
** *C. krusei* **	01	01	-	01	-	-	-	-	-	-	-	-
**Number of *Candida* colonies**	11.619 ± 2.72	5.238 ± 2.40	0.000	9 ± 2.70	4.25 ± 2.35	0.000	10 ± 2.74	5.333 ± 2.37	0.001	10.666 ± 2.70	5.111 ± 2.38	0.000

**Table 7 healthcare-10-01507-t007:** Association between the class of cytology and type of *Candida*.

Class of Cytology	*C. albicans*			*C. tropicalis*			*C. glabrata*			*C. krusei*			Total Class of Cytology		
	Pre	Post	Control	Pre	Post	Control	Pre	Post	Control	Pre	Post	Control	Pre	Post	Control
**I**	-	08	44	02	05	11	-	-	04	-	-	01	02	13	60
**II**	08	19	-	17	05	-	04	-	-	02	01	-	31	25	-
**III**	04	15	-	18	06	-	05	01	-	-	-	-	27	22	-
**IV**	-	-	-	-		-	-	-	-	-	-	-			
**Total (Type of Candida)**	12	42	44	37	16	11	9	01	04	02	01	01			

**Table 8 healthcare-10-01507-t008:** Association between the type of tobacco and type of *Candida*.

Type of Tobacco	*C. albicans*		*C. tropicalis*		*C. glabrata*		*C. krusei*		Total
	Pre	Post	Pre	Post	Pre	Post	Pre	Post	
**Smokeless**	08	24	18	7	4	-	1	-	31
**Smoking**	01	07	5	2	3	-	-	-	9
**Smokeless and Smoking**	03	11	14	7	2	1	1	1	20

## Data Availability

Not applicable.

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
