# Peer review of "Assessing the Effect of Curcumin on the Oral Mucosal Cytomorphometry and Candidal Species Specificity in Tobacco Users: A Pilot Study"

_healthcare, 2022, doi:10.3390/healthcare10081507_

Round 1

Reviewer 1 Report

The study I think should include pilot study in the title due to limited sample size and limited duration of study. 

In the method, Sample size was 145 participants …might need explanation why this sample size was selected. Was there a time frame for patient enrollment in the study? 

In the method, In inclusion criteria, “History of tobacco and related substance use for a minimum of 6 months” is this an acceptable time period for oral lesions to form and develop candida..please explain and justify with evidence. 

In the method, Please provide detailed what were the oral examination done

and oral hygiene instructions given. Was the history of habits that the patients were asked in English? was it  translated? And if translated ? was it validated? Please provide details and add to manuscript as supplemental if possible.  

In the method, 

 “All smears of both groups were reviewed independently by three examiners “ who were the examiners?

In the results, was there data the researcher compared between conventional way of candida identification to prove its superiority to conventional Dextose Agar method.

This point was not explained well in discussion section. 

In the result, the study did not show what type of tobacco was used in the study population. And justify if it has any effect on candida type, shift seen in the results. 

In the statistics, the sample size diagram has no legend. 

In discussion, the term “non-habit group” need to be defined in the methods and decide to use this or study group consistently throughout the manuscript. 

In discussion, there was no enough explanation about candida type variation and shift seen after treatment. 

Author Response

Reviewer 1:   Comments

Corrections

1.  

The study I think should include pilot study in the title due to limited sample size and limited duration of study. 

As per suggestions, word as a pilot study included in the title.

2.  

In the method, Sample size was 145 participants …might need explanation why this sample size was selected. Was there a time frame for patient enrollment in the study? 

Sample size was calculated based on number of patients visiting out patient department of K.M.Shah Dental College and Hospital and considering two months duration for study.

The grant for the research was provided by STS-ICMR and in accordance to the guidelines provided by STS-ICMR the study had to be completed within 2 months, therefore keeping the number of OPD in mind, the following sample size was decided with the help of the  formula as shown in manuscript.

3.

In the method, In inclusion criteria, “History of tobacco and related substance use for a minimum of 6 months” is this an acceptable time period for oral lesions to form and develop candida..please explain and justify with evidence. 

Prolonged exposure as tobacco in chewable form especially where tobacco is kept in contact with the mucosa for a longer time leading to leaching & concentration of carcinogens at a localized area increases the chances of developing lesion As reported by Garg et al, in these cases, longer individual chewing cycle even at short duration may be hazardous. Lesion may also develop as a protective mechanism for  localized carcinogen action.

Tobacco components increase the production of reactive oxygen species, cell turnover, collagen synthesis and also alters DNA damage.

Also, depending upon the damage occurred, clinical presentation varies from greyish-white ill -demarcated lesions, to ulceration. Recent research by Halboub et al 2020 & Monika et al 2020 have reported that various types of smokeless tobacco harbor bacteria that play role in carcinogenicity.

Mechanical & chemical irritation from smokeless tobacco also induces melanin pigmentation.

Miller et al reported that, smokeless tobacco lesion should be treated by habit discontinuity which shows resolution within 6 weeks to 6 months, so minimum of 6 months duration of  history of tobacco habit was considered as inclusion criteria for study.

4.

In the method, Please provide detailed what were the oral examination done and oral hygiene instructions given. Was the history of habits that the patients were asked in English? was it  translated? And if translated ? was it validated? Please provide details and add to manuscript as supplemental if possible.  

For present study, translation, back translation in the local language of Gujarati & Hindi and validation of the same was mandatory protocol for ethics committee approval, herewith we are attaching the copy of  case history proforma for history of habits, for reference.

Extra oral examination:

Face and Head was assessed to find abnormal findings like symmetry, swelling or dis-colouration

Neck was palpated to assess major lymph nodes for any lumps, swelling and tenderness

Intra oral examination :

Examination was done with the help of mouth mirror and probe. Lips, Palate, Tongue, Floor of Mouth, Buccal mucosa, Labial mucosa and Fauces to check for Dis-colouration, texture, keratinisation, swelling, consistency or any other abnormality such as fibrous bands.

Following were the oral hygiene instructions

1. Brush teeth twice a day with soft bristled toothbrush, once in morning and once at night

2. Rinse and swish with water after every meal

3. Floss once daily

4. Brush the tongue

5.

In the method,  “All smears of both groups were reviewed independently by three examiners “ who were the examiners?

All smears were reviewed independently by all three faculties  which included both professors and assistant professor, from the department of oral pathology.

6.  

In the results, was there data the researcher compared between conventional way of candida identification to prove its superiority to conventional Dextose Agar method.

This point was not explained well in discussion section. 

Candida identification was done by culture done on conventional Dextrose Agar  and was also confirmed by gram stain. No other data was obtained by comparison with any other conventional way of candida identification.

Limitation of the study has considered this point.

The other conventional methods of candida culture other than SDA such as germ tube & chymadospore formation were time consumingso not compared with cheromagar. We had used SDA which is considered as gold standard and whose sensitivity is considered equal to conventional methods.[12]

7.  

In the result, the study did not show what type of tobacco was used in the study population. And justify if it has any effect on candida type, shift seen in the results

Table 7. Association between the type of tobacco and type of Candida:.

Type of Tobacco

C.albicans

C.tropicalis

C.glabrata

C.krusei

Total

Pre

Post

Pre

Post

Pre

Post

Pre

Post

Smokeless

08

24

18

7

4

-

1

-

31

Smoking

01

07

5

2

3

-

-

-

9

Smokeless & Smoking

03

11

14

7

2

1

1

1

20

Among 60 participants of study group, 31 were smokeless tobacco chewers, 20 had both smokeless                                and                                    smoking habit whereas 09 participants were smokers.

In our study, candida tropicalis was found more in smokeless tobacco (18) consuming group followed by candida albicans(8), candida glabrata(4) and candida krusei(1). After Intervention, candida albicans(24) was more prevalent followed by candida tropicalis(7)

In smoking tobacco consuming group, candida tropicalis was found prevalent (5) followed by candida glabrata(3) and candida albicans(1). After Intervention, candida albicans(7) was more prevalent followed by candida tropicalis(2)

In the study group, participants consuming both smokeless and smoking type of tobacco had candida tropicalis prevalent(14) followed by candida albicans(3), candida glabrata(2) and candida krusei(1). After Intervention, candida albicans(11) was more prevalent followed by candida tropicalis(7) , candida glabrata(1) and candida krusei(1)

In all three study groups we found detection of candida tropicalis commonly as compared to other species of candida. And in all the study groups of habits irrespective of type of tobacco habit, post intervention, there was shift of candida tropicalis to candida albicans.

8.

In the statistics, the sample size diagram has no legend. 

Added this point in statistical analysis as well as in flow diagram

Statistical Analysis:

Sample size was calculated as per number of patients visiting out patient department of K.M.Shah Dental College and Hospital and considering two months duration for study.

9.

In discussion, the term “non-habit group” need to be defined in the methods and decide to use this or study group consistently throughout the manuscript. 

In discussion as well as throughout the manuscript, term control group for the participants of no habit has been preferred to use.

10

In discussion, there was no enough explanation about candida type variation and shift seen after treatment.

 The presence of Candida tropicalis has been reported by Keten et a [46] & Krogh et al [49]. The C. tropicalis strain has been reported similarly in studies by Keten et al [46]. Agree for the occurrence of Candida tropicalis is a common finding in tobacco users whereas Candida albicans in non tobacco users.

Thus, in our study preintervention, Candida tropicalis was found whereas post intervention there was shift of Candida tropicalis to that of Candida albicans in habit group irrespective of type of habit of smokeless tobacco or smoking form of tobacco.

Candida albicans( which is also a commensal of oral cavity) in habit group irrespective of type of habit of smokeless tobacco or smoking form of tobacco.

In our study, there was reduction in the number of candida colonies after the intervention with curcumin depicting antifungal property of curcumin. It has been seen that       curcumin has antifungal effect on candida tropicalis, candida albicans, candida glabarata and candida krusei            [59,60]

Reviewer 2 Report

Dear Authors

the manuscript is interesting and well written. I suggest some changes to improve the manuscript:

1) specify that the manuscript is a prospective study 

2) modify the flow-chart diagram (all the diagrams must be in the same style)

3) Provide final photo or image of the patients lesions after the curcumina treatments

4) Improve same sentences in English language that are too long or unclear. 

Author Response

Reviewer 2: Comments:

1.

specify that the manuscript is a prospective study 

The study was a pilot & prospective study with intervention by curcumin gel and has been specified as per suggestions.

2.

modify the flow-chart diagram (all the diagrams must be in the same style)

Flow chart diagrams are modified as suggested.

3.

Provide final photo or image of the patients lesions after the curcumina treatments

Pre and Post intervention photographs are added.

4.

Improve same sentences in English language that are too long or unclear. 

Modifications done.